# Changing diagnostic criteria for gestational diabetes (CDC4G) in Sweden: A stepped wedge cluster randomised trial

Maryam de Brun[1], Anders Magnuson[2], Scott Montgomery[2,3,4], Snehal Patil[2], David Simmons[5,6], Kerstin Berntorp[7], Stefan Jansson[8], Ulla-Britt Wennerholm[9], Anna-Karin Wikström[10], Helen Strevens[11], Fredrik Ahlsson[10], Verena Sengpiel[9], Erik Schwarcz[12], Elisabeth Storck-Lindholm[13], Martina Persson[14], Kerstin Petersson[15], Linda Ryen[8], Carina Ursing[16], Karin Hildén[1], Helena Backman[1]*

1 Department of Obstetrics and Gynaecology, Faculty of Medicine and Health, Örebro University, Örebro, Sweden, 2 Clinical Epidemiology and Biostatistics, School of Medical Sciences, Faculty of Medicine and Health, Örebro University, Örebro, Sweden, 3 Clinical Epidemiology Division, Department of Medicine, Solna, Karolinska Institutet, Stockholm, Stockholm, Sweden, 4 Department of Epidemiology and Public Health, University College London, London, United Kingdom, 5 School of Medical Science, Faculty of Medicine and Health, Örebro University, Örebro, Sweden, 6 Macarthur Clinical School, Western Sydney University, Campbelltown, Australia, 7 Genetics and Diabetes Research Unit, Department of Clinical Sciences Malmö, Lund University, Malmö, Sweden, 8 University Health Care Research Centre, Faculty of Medicine and Health, Örebro University, Örebro, Sweden, 9 Department of Obstetrics and Gynaecology, Institute of Clinical Sciences, Sahlgrenska Academy, University of Gothenburg and Region Västra Götaland, Sahlgrenska University Hospital, Department of Obstetrics and Gynecology, Gothenburg, Sweden, 10 Department of Women's and Children's Health, Uppsala University; Uppsala University Hospital, Uppsala, Sweden, 11 Department of Obstetrics and Gynaecology, Skåne University Hospital, Department of Clinical Sciences Lund, Lund University, Lund, Sweden, 12 Department of Medicine, Faculty of Medicine and Health, Örebro University, Örebro, Sweden, 13 Department of Obstetrics and Gynaecology Södersjukhuset, Karolinska Institute, Solna, Sweden, 14 Department of Clinical Science and Education Karolinska Institute, Department of Medicine, Clinical Epidemiology Karolinska Institutet and Sachsska Childrens'and Youth Hospital Stockholm, Stockholm, Sweden, 15 Department of Obstetrics and Gynaecology Södersjukhuset, Umeå University, Umeå, Sweden, 16 Department of Endocrinology and Diabetology, Södersjukhuset, Stockholm, Sweden

* Helena.backman@oru.se, Helena.backman@regionorebrolan.se

**Data Availability Statement:** The datasets generated and/or analysed during the current study

## Abstract

### Background

The World Health Organisation (WHO) 2013 diagnostic criteria for gestational diabetes mellitus (GDM) has been criticised due to the limited evidence of benefits on pregnancy outcomes in different populations when switching from previously higher glycemic thresholds to the lower WHO-2013 diagnostic criteria. The aim of this study was to determine whether the switch from previous Swedish (SWE-GDM) to the WHO-2013 GDM criteria in Sweden following risk factor-based screening improves pregnancy outcomes.

### Methods and findings

A stepped wedge cluster randomised trial was performed between January 1 and December 31, 2018 in 11 clusters (17 delivery units) across Sweden, including all pregnancies under

are not publicly available due current Swedish ethical legislation and European union GDPR act, but are available from the organisation on reasonable request, if appropriate permits are obtained from adequate authorities. Requests should be directed to: "foudatauttag@regionorebrolan.se".

**Funding:** Swedish Research Council (https://www.vr.se/english.html) (HB), 2018-00470, ALF Funding Region Örebro County (HB) OLL-930268, The Swedish state under the agreement between the Swedish government and the county councils, the ALF-agreement , (VS), GBG-823211, ALFGBG-932692, Nyckelfonden,Region Örebro County, (HB), OLL-597601, Region Örebro County Research committee (HB), OLL-693551, OLL-786911, Regional Research committee Uppsala-Örebro (HB), RFR-749241, Stiftelsen Mary von Sydows, född Wijk, donation fund, (VS), numbers 1017, 4917, 2618, and 3718), Clinical therapy research, Region Stockholm County, The Centre of Clinical Research, (ESL), Västmanland County Council, (MdB), LTV-966501, Research Funds of Skåne University Hospital and the Skåne County Council Research and Development Foundation (KB), REGSKANE-622891. The funders had no role in study design, data collection and analysis, decision to publish, or preparation of the manuscript.

**Competing interests:** The authors have declared that no competing interests exist.

**Abbreviations:** aRR, adjusted risk ratio; BMI, body mass index; CI, confidence interval; GDM, gestational diabetes mellitus; ITT, intention to treat; LGA, large for gestational age; mITT, modified intention to treat; mPP, modified per-protocol population; OGTT, oral glucose tolerance test; RCT, randomised controlled trial; RR, risk ratio; SAP, statistical analysis plan; SD, standard deviation; SW-CRT, stepped wedge cluster randomised controlled trial; WHO, World Health Organisation.

care and excluding preexisting diabetes, gastric bypass surgery, or multifetal pregnancies from the analysis.

After implementation of uniform clinical and laboratory guidelines, a number of clusters were randomised to intervention (switch to WHO-2013 GDM criteria) each month from February to November 2018. The primary outcome was large for gestational age (LGA, defined as birth weight >90th percentile). Other secondary and prespecified outcomes included maternal and neonatal birth complications. Primary analysis was by modified intention to treat (mITT), excluding 3 clusters that were randomised before study start but were unable to implement the intervention. Prespecified subgroup analysis was undertaken among those discordant for the definition of GDM. Multilevel mixed regression models were used to compare outcome LGA between WHO-2013 and SWE-GDM groups adjusted for clusters, time periods, and potential confounders. Multiple imputation was used for missing potential confounding variables.

In the mITT analysis, 47 080 pregnancies were included with 6 882 (14.6%) oral glucose tolerance tests (OGTTs) performed. The GDM prevalence increased from 595/22 797 (2.6%) to 1 591/24 283 (6.6%) after the intervention. In the mITT population, the switch was associated with no change in primary outcome LGA (2 790/24 209 (11.5%) versus 2 584/22 707 (11.4%)) producing an adjusted risk ratio (aRR) of 0.97 (95% confidence interval 0.91 to 1.02, $p = 0.26$).

In the subgroup, the prevalence of LGA was 273/956 (28.8%) before and 278/1 239 (22.5%) after the switch, aRR 0.87 (95% CI 0.75 to 1.01, $p = 0.076$). No serious events were reported. Potential limitations of this trial are mainly due to the trial design, including failure to adhere to guidelines within and between the clusters and influences of unidentified temporal variations.

## Conclusions

In this study, implementing the WHO-2013 criteria in Sweden with risk factor-based screening did not significantly reduce LGA prevalence defined as birth weight >90th percentile, in the total population, or in the subgroup discordant for the definition of GDM. Future studies are needed to evaluate the effects of treating different glucose thresholds during pregnancy in different populations, with different screening strategies and clinical management guidelines, to optimise women's and children's health in the short and long term.

## Trial registration

The trial is registered with ISRCTN (41918550).

---

### Author summary

#### Why was this study done?

- The implementation of the World Health Organisation (WHO)-2013 diagnostic criteria for gestational diabetes mellitus (GDM) have been challenged due to the limited evidence of benefits on pregnancy outcomes in different populations by switching from

former higher plasma glucose diagnostic cutoffs to the lower plasma glucose WHO-2013 diagnostic criteria.

- Screening, laboratory methods, and diagnostic criteria for GDM vary throughout the world and there is limited randomised controlled trial (RCT) evidence on the effects of switching to WHO 2013 diagnostic criteria for GDM.

- The Swedish National Board of Health and Welfare introduced new guidelines for GDM in 2015 and the aim was to evaluate if the switch in a real-world setting improved pregnancy outcomes.

**What did the researchers do and find?**

- A stepped wedge randomised trial was performed during 2018 which included nearly half of all pregnancies in Sweden that year ($n$ = 47 080). Since risk factor screening was used, analysis was conducted in all pregnancies (modified intention to treat (mITT)) as well as in a subgroup affected by the switch.

- There was no reduction in the main outcome large for gestational age (LGA) (>90th birth weight percentile) in the mITT population or in the subgroup of women affected by the switch.

**What do these findings mean?**

- These findings indicate that the effect of treatment may differ using lower compared to higher plasma glucose diagnostic cutoffs for GDM depending on whether risk factor based screening or universal screening is used.

- The study findings highlight the importance of also reporting treatment effects on high absolute birth weight besides the LGA 90th percentile, since absolute high birth weight most likely results in associated adverse pregnancy outcomes.

- Limitations of this trial are mainly due to the trial design, including failure to adhere to guidelines within and between the clusters and influences of unidentified temporal variations.

- Future studies need to evaluate long-term effects on women's and children's health after diagnosing and treating lower levels of hyperglycemia during pregnancy.

## Introduction

Gestational diabetes mellitus (GDM) is the most common medical complication of pregnancy with a growing prevalence globally, to a large extent due to the increase in overweight and obesity [1]. Hyperglycemia during pregnancy is associated with complications for mother and child during pregnancy and delivery, but also associated with raised risks of later type 2 diabetes and cardiovascular disease for the mother. For the child, there is emerging evidence about

risks for future obesity and associated comorbidity [2–4]. The clinical controversy about what glycemic thresholds to diagnose and treat GDM relates to the balance between sufficient evidence of health improvements in different populations and increased workload with associated costs [5–7]. To progress towards a universal standard approach to GDM diagnosis, the World Health Organisation (WHO) adopted the International Association of the Diabetes and Pregnancy Study Group's diagnostic criteria in 2013 [1]. These WHO-2013 criteria, with an increase in the number of women diagnosed [8], are based on risk for adverse pregnancy outcomes, in contrast to older criteria relating to maternal risk of developing type 2 diabetes postpartum. It has been unclear whether treatment using these WHO-2013 criteria improves outcomes for the mother or the child. The few randomised trials evaluating the effect of GDM treatment have been performed in different populations, with varying screening strategies and comparing different diagnostic criteria for GDM [5–7].

In Sweden, screening for GDM has mostly been performed based on clinical risk factors and repeated random plasma glucose measurements. Diagnostic criteria have been mainly those of overt diabetes (fasting ≥7.0 mmol/L, 2-h value 8.9 to ≥11.1 mmol/L) [9], resulting in a low prevalence of GDM compared with other countries [10]. In 2015, the Swedish National Board of Health and Welfare adopted the WHO-2013 criteria. A national stepped wedge cluster randomised controlled trial (SW-CRT) was considered a pragmatic approach to test whether a reduction in adverse neonatal and/or maternal outcomes could be detected following the implementation of the WHO-2013 criteria in Sweden in a real-world setting [9].

The primary aim of the Changing Diagnostic Criteria for Gestational Diabetes (CDC4G) trial was to evaluate whether implementation of the WHO-2013 criteria leads to a reduction in large for gestational age (LGA, 90th percentile) infants, and the secondary aim was to evaluate possible reduction of other adverse neonatal and maternal outcomes.

## Methods

This study is reported as per the Consolidated Standards of Reporting Trials extension for SW-CRT (S1 CONSORT Checklist) [11]. The details of the trial methodology are described in original ethics project plan (S1 Appendix), study protocol paper as well as in published SAP with corrections (S2 Appendix) [12]).

### Study design and participants

The CDC4G trial was a national prospective, unblinded, open SW-CRT with concealed allocation of the switch from the former Swedish GDM criteria (SWE-GDM criteria) to the WHO-2013 criteria during 2018. Eleven clusters were defined from the 17 participating Swedish delivery units. Delivery units in Stockholm were considered as 1 cluster since guidelines; surveillance, diagnosis, and treatment of GDM were uniform and women might change caregivers (midwifes and delivery units) during pregnancy. Therefore, a woman could not be identified as belonging to just 1 delivery unit, as described in S1 Table. During the preparation phase (September 1 to December 31, 2017), all clusters agreed to shift to a uniform approach to GDM management, including venous oral glucose tolerance test (OGTT) blood sampling, obstetric surveillance, and GDM treatment [13].

All clusters agreed not to change screening methods (S2 Table) or clinical management guidelines (S3–S5 Tables) during the trial period. All women under the care of a participating clinic during the trial period were included. Predefined exclusion criteria according to the statistical analysis plan (SAP S2 Appendix) [12] were: clusters that did not adhere to the trial protocol procedures and women not eligible for OGTT such as preexisting diabetes and previous gastric bypass surgery (OGTT contraindicated). Multifetal pregnancies were excluded because

fetal growth and pregnancy complications are not comparable to singleton pregnancies. The study period was from January 1, 2018 to December 31, 2018 and the OGTT date or gestational week 28+0 (i.e., before or after the planned intervention date) determined which period each woman was allocated to (S6 Table) with last birth in August 2019.

The primary analysis was performed according to the intention-to-treat principle, consisting of all eligible pregnancies in the randomised clusters. The intention to treat (ITT) population defined in the SAP is labelled as the modified intention to treat (mITT) population in this manuscript and includes only the centres that fulfilled the eligibility criteria (introducing venous OGTT for diagnosis of GDM) [14]. The results for the complete ITT population were calculated, including all clusters across Sweden who agreed to participate in the CDC4G trial (S16–S18 Tables). The same analyses were performed in a prespecified subgroup of pregnancies that would have been untreated before randomisation and treated after the switch based on the fasting and 2-h plasma glucose cut off values; i.e., discordant for definition of GDM. The glucose values were between SWE-GDM and WHO-2013 criteria: fasting plasma glucose 5.1 to 6.9 and/or 2-h plasma glucose 8.5–8.8/8.9/9.9 mmol/L according to definitions in S2 Table. One-hour values were not recorded before the switch due to laboratory and clinical regulatory issues making it impossible to blind the 1-h value in clinical practice for patients and caregivers. The modified per-protocol population (mPP) consists of all pregnancies in the randomised clusters having commenced the trial until exclusion due to protocol violation.

The trial was approved by the Uppsala-Örebro regional Ethical Review Board (2016/487) and by the Swedish Ethical Review Authority (2019/02148, 2020/02856, 2021/02055, 2021-03405). The Uppsala-Örebro regional Ethical Review Board agreed that individual patient consent was not required. According to Swedish law, women always have the right to change clinics, refuse any aspect of medical care, and opt out of the Swedish Pregnancy Register.

## Randomisation and masking

A cluster randomisation of allocation ratio of 1:1 stratified by centre size in 2 strata by size was conducted. The 2 largest trial clusters (Gothenburg and Stockholm) were paired in 1 stratum and randomised to change GDM criteria in June and August of 2018, respectively. The second stratum comprised the remaining 9 clusters randomised to the intervention (WHO-2013 criteria), 1 cluster each month (period), from February to July and September to November 2018 (Fig 1). The randomisation allocation was performed using computer-generated random allocation sequences provided by the trial statistician. The randomisation was performed in November 2017 and masked from the participating clusters and steering group until 2 months prior to the start of intervention for each cluster. Two months were the estimated preparation time needed for each cluster before switching.

## Procedures

While the SWE-GDM criteria consistently used fasting plasma glucose criterion of ≥7.0 mmol/L, the 2-h plasma glucose threshold varied between 8.9 and 11.1 mmol/L across clusters, as described in S2 Table. The switch included moving from two-point fasting and 2-h to a three-point venous OGTT with fasting plasma glucose, 1-h and/or 2-h diagnostic thresholds of ≥5.1, ≥10.0, and ≥8.5 mmol/L at cluster level (Fig 1). The screening method remained unchanged (S2 Table).

The trial statistician provided information on the cluster randomisation date to the trial coordinator, who in turn informed the relevant cluster principal investigator 2 months prior to transition to the WHO-2013 criteria, to ensure complete cluster preparation before the date

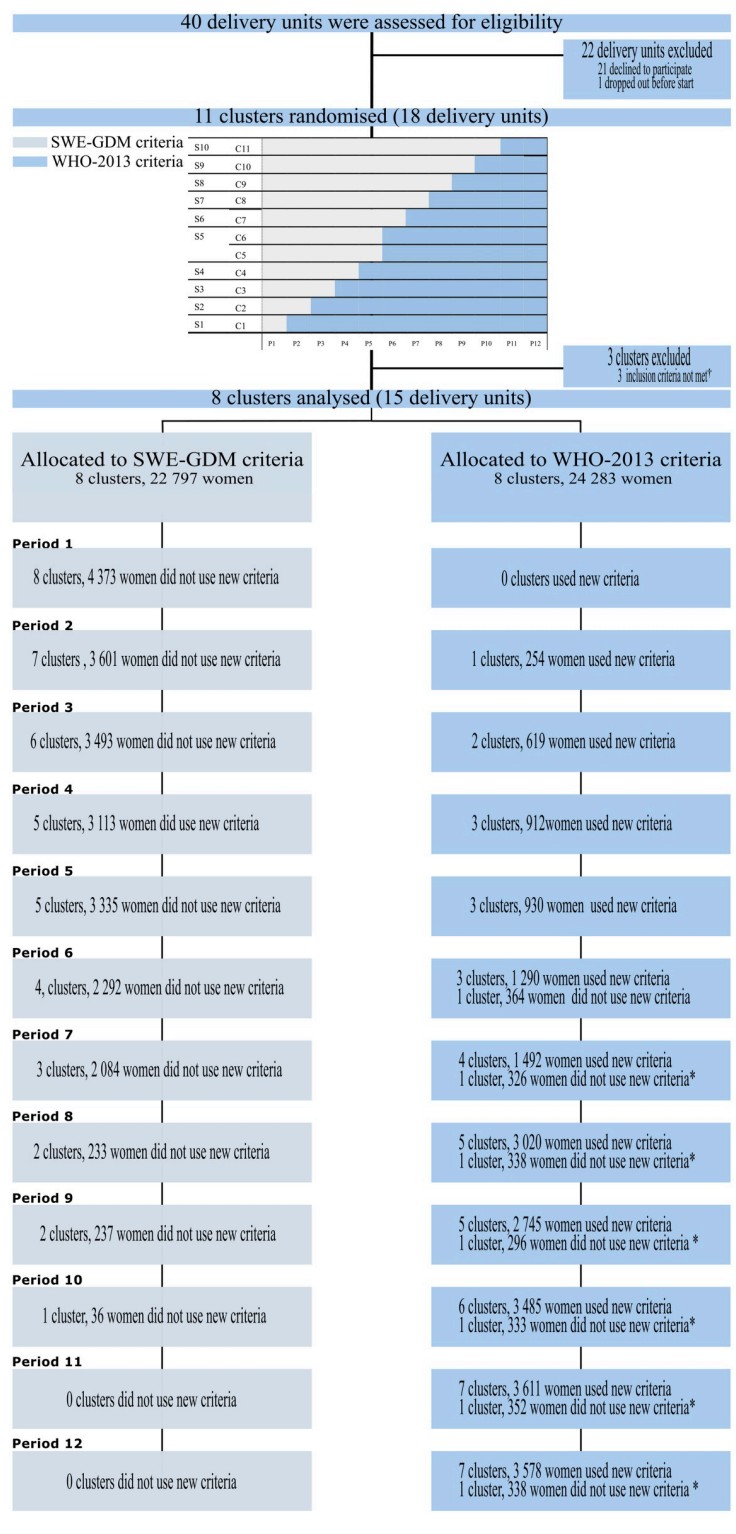

**Fig 1. Trial profile.**

of the switch. Checklists completed by local principal investigators assessed adherence to trial protocol, GDM prevalence, and serious adverse events monthly.

Serious adverse events were defined as maternal death (death of mother included in the trial during the trial period), severe maternal hypoglycemia (low plasma glucose levels resulting in cognitive impairment that requires assistance from another person to treat), and/or lactic acidosis in metformin-treated women and were reported to the data and safety monitoring board.

### Data collection

Data were collected on all pregnancies in clinical care at the included clusters in the study between 2017 and 2019 using national registers and electronic case report forms after the end of the trial period. Data quality is described in supplementary S7 Table [9,13]. No exclusions or inclusions were made at study start or during the data collection phase. Health and quality registers provides standardised medical information on all pregnancies, with a coverage of >95% [15]. Pregnancy outcome data from the Swedish pregnancy quality register is available online for all clinics for health care quality surveillance. The Swedish National Board of Health and Welfare completed merging of data using personal identification numbers after data collection was completed. This was done after their review of ethical permission and according to Swedish laws and regulations. The pseudonymised files containing data on pregnancies from 2017 to 2019 were received between June 20, 2022 and September 30, 2022 (delayed due to the COVID pandemic). Data management and validity control for the data files was performed thereafter. Exclusion for predefined criteria was carried out at the analysis stage after the SAP was finalised.

The study cohort and predefined outcomes were described in the SAP published on October 28, 2022 with clarifications decided by the steering group January 27, 2023 and published on May 12, 2023 at ISRCTN, which added 3 further exploratory outcome variables [12] (S2 Appendix). The study statistician received the data set with primary outcome on November 29, 2022. Analysis of all outcomes was performed after the SAP was published, with the exception of the 3 exploratory variables that were added to the data set after October 28, 2022 and analysed after correction of the SAP.

### Outcomes

The primary outcome was LGA, defined as birth weight >90th percentile in the Swedish reference population according to Marsál and colleagues [16] corrected for gestational age and sex [9,13]. All reported outcomes were predefined according to the SAP which was completed before analysis [12]. Some changes were made to the secondary outcome measures after their original description in ethics application (S1 Appendix). Final decisions on outcomes were decided based on either the study protocol paper, core outcome set publication for GDM [17], and/or outcomes reported in major studies in the research field for comparison reasons. Not all outcomes in the SAP are included in this first paper but will be included in further papers.

### Secondary neonatal outcomes

Neonatal outcomes included a composite measure created through investigator consensus and was slightly differently defined from the protocol in the original ethical application but reflects severe morbidity with valid variables from the registers used (respiratory distress (at least 4 hours' respiratory support with supplemental oxygen, continuous positive airway pressure, and/or intermittent positive pressure ventilation in the first 24 h after delivery), birth trauma (spinal cord injury, peripheral nerve injury/brachial plexus, basal skull fracture or depressed

skull fracture, clavicular fracture, long bone fracture including humerus, radius, ulna, femur, tibia or fibula, cranial hemorrhage including subdural or intracerebral of any kind confirmed by cranial ultrasound, computerised tomography scan, or magnetic resonance imaging), stillbirth (fetal death at ≥22 + 0 weeks' gestation), death of a neonate (≤28 day), and/or need for therapeutic cooling). The individual components of the composite outcome are reported separately. Other secondary neonatal outcomes included preterm birth (<37+0 weeks), small for gestational age (birth weight <10th percentile according to Marsál and colleagues reference population [16] corrected for gestational age and sex), 5 min Apgar score <4, metabolic acidosis (pH <7.05 and base excess >12 mmol/L in umbilical artery or pH <7.00 in umbilical artery), admittance to neonatal intensive care unit (days) (>24 h), hypoxic ischaemic encephalopathy II-III, meconium aspiration syndrome, mechanical ventilation, plasma glucose in infants <2.6 mmol/L, hypoglycemia needing intravenous therapy.

### Exploratory neonatal outcomes

Exploratory neonatal outcomes included macrosomia (≥4 500 g), severe LGA (>2 standard deviation (SD) [16]), severe small for gestational age (<2 SD [16]), birth length (cm), birth weight (g), and gestational age (days).

### Secondary maternal outcomes

Secondary maternal outcomes included a composite variable of adverse outcomes (shoulder dystocia, perineal trauma (grades III and IV), and postpartum hemorrhage (≥1 000 ml)). The individual components of the composite outcomes are reported separately. Other maternal outcomes included GDM treatment during pregnancy (diet, metformin, insulin), gestational hypertension (new-onset blood pressure ≥140/90 mmHg, measured twice with at least 4-h interval, after gestational week 20), preeclampsia (gestational hypertension combined with new-onset significant proteinuria after gestational week 20), gestational weight gain, cesarean section (elective, emergency), instrumental delivery, induction of labour, breastfeeding at discharge, self-reported health during and after pregnancy (very good, good, neither good nor bad, bad, very bad), and satisfaction with childbirth measured at discharge (1 indicates worst experience and 10 best experience) [18].

### Exploratory maternal outcome

Maternal death (up to 42 days after delivery with deaths due to accidents excluded).

### Sample size and statistical analysis

In the sample size calculation for the main LGA outcome, 11 clusters were planned with an assumed intracluster correlation of 0.0026. A minimum sample size of 23 958 pregnant women per trial group was estimated to have 90% statistical power to detect an absolute reduction in LGA by 1.5% on a population level (from 10.0 to 8.5%) at a 5% significance level [19]. With 80% power 35 112 pregnancies (17 556 in each group) would detect the same difference. The intracluster correlation was estimated from the variation in LGA prevalence in the year 2012 between clusters, which varied from 7.7% to 13.3%.

The recommendations for analysis of SW-CRT were followed [20]. Binary outcomes, including the primary outcome, were analysed using multilevel mixed Poisson regression with robust standard errors to compare the WHO-2013 and SWE-GDM criteria groups with clusters as a random factor and calendar time period (January to March, April to June, July to September, and October to December) as a fixed factor. Poisson regression gave relative risk

ratios (RRs) with 95% confidence intervals (CIs) as effect measures. Categorical outcomes with more than 2 categories were analysed in the same way using mixed multinomial regression, continuous outcomes using linear mixed regression, and count data outcomes using negative binomial mixed regression. Adjustment was made for the potential confounding variables for associations with LGA: maternal age, chronic hypertension, smoking habits and Swedish snuff habits in early pregnancy, country of birth classified according to the International Diabetes Federation Diabetes Atlas, and parity. Maternal age was modelled by linear, squared, and cubic terms. Body mass index (BMI) was not adjusted for since screening for GDM was mainly undertaken using BMI, and this was therefore considered to be an over adjustment. If the regression did not converge due to a sparse number of outcomes, the potential confounding variables were collapsed into fewer categories or restricted analysis was performed. "Not applicable" was reported in the tables if the mixed model did not converge due to a limited number of outcome events. Multiple imputation using chained equations was used to impute 10 data sets for the missing potential confounding variables. All above explanatory variables together with mother's BMI at first visit, education level and clusters were used in the imputation model. The STATA command *mi estimate* was used to adjust for variability between the 10 imputed data sets according to the combination rules of Rubin [21]. Due to large proportion of missing outcome data for breastfeeding at discharge and self-reported health during and after pregnancy, multiple imputation was also used for these outcomes as sensitivity analysis.

No corrections were made for multiple comparisons among the prespecified outcomes.

An independent statistician blinded to the studied groups used STATA release 17.0 for all statistical analysis. The trial was registered with ISRCTN (41918550) including the SAP before the analyses were started [12,13,22].

## Results

Between January 1 and December 31, 2018, 17 delivery units (11 clusters, 58 383 pregnancies) across Sweden agreed to participate in the CDC4G trial (ITT population). Three clusters (clusters 4, 5, and 9), constituting 11 303 (19.4%) pregnancies, were randomised but excluded before intervention, as they inappropriately continued to use one-step universal capillary OGTT results for screening, diagnosis, and treatment throughout the trial period and thus did not introduce the defined intervention, resulting in the mITT that included 47 080 pregnancies. One cluster (cluster 2), switched back to SWE-GDM diagnostic criteria during periods 6 to 12 due to an unmanageable workload, resulting in exclusion of 2 437 pregnancies. Another 83 pregnancies were excluded due to capillary OGTT sampling in other clusters. After these exclusions, the modified per protocol population consisted of 44 643 pregnancies.

The mITT population included 24 283 pregnancies in the WHO-2013 criteria period and 22 797 in the SWE-GDM criteria period (Fig 1 and detailed in S1 Fig). There were 6 882 OGTTs (14.6%) with 3 747 (15.4%) OGTTs using the WHO-2013 criteria and 3 135 (13.8%) using the SWE-GDM criteria. In the mPP population, there were 21 886 pregnancies using the WHO-2013 criteria and 22 757 using the SWE-GDM criteria. In the subgroup discordant for definition of GDM, there were 1 239 pregnancies in the WHO-2013 group and 956 pregnancies in the SWE-GDM group.

In the mITT population, the women in the WHO-2013 criteria group had higher BMI and parity, were more likely to smoke, were less likely to be Swedish-born, and had a lower education level compared to the SWE-GDM group (Table 1).

Women in the subgroup discordant for definition of GDM were younger, had lower prevalence of chronic hypertension, and were less likely to be Swedish-born, had lower education level, and higher mean OGTT fasting, 1- and 2-h glucose values during the WHO-2013 criteria

**Table 1. Baseline characteristics of the modified ITT population and the subgroup discordant for definition of GDM*.**

| Maternal characteristics | mITT population | | | | Subgroup discordant for definition of GDM* | | | |
| --- | --- | --- | --- | --- | --- | --- | --- | --- |
| | SWE-GDM criteria ($n$ = 22 797) | | WHO-2013 criteria ($n$ = 24 283) | | SWE-GDM criteria ($n$ = 956) | | WHO-2013 criteria ($n$ = 1 239) | |
| | $n$ | | $n$ | | $n$ | | $n$ | |
| Age at childbirth, years | 22 797 | 31.6 (28–35) | 24 283 | 31.3 (28–35) | 956 | 32.8 (29–36) | 1 239 | 32.8 (29–36) |
| Body height at first visit, cm | 22 079 | 166 (162–170) | 23 495 | 166 (161–170) | 933 | 166 (162–170) | 1 222 | 165 (160–170) |
| Body weight at first visit, kg | 21 655 | 66 (59–75) | 23 041 | 66 (59–75) | 922 | 86 (70–100) | 1 204 | 83 (69–98) |
| BMI at first visit, kg/m² | 21 591 | 23.7 (21.5–27.0) | 23 041 | 23.9 (21.6–27.3) | 922 | 30.9 (25.6–35.9) | 1 202 | 30.4 (25.6–35.8) |
| Underweight (<18.5) | | 547 (2.5) | | 620 (2.7) | | 5 (0.5) | | 7 (0.6) |
| Normal (18.5–24.9) | | 12 705 (58.8) | | 13 157 (57.1) | | 192 (20.8) | | 257 (21.4) |
| Overweight (25.0–29.9) | | 5 458 (25.3) | | 5 908 (25.6) | | 212 (23.0) | | 300 (25.0) |
| Obesity class I (30.0–34.9) | | 2 039 (9.4) | | 2 289 (9.9) | | 211 (22.9) | | 273 (22.7) |
| Obesity class II (35.0–39.9) | | 620 (2.9) | | 760 (3.3) | | 219 (23.8) | | 254 (21.1) |
| Obesity class III (≥40.0) | | 222 (1.0) | | 307 (1.3) | | 83 (9.0) | | 111 (9.2) |
| Parity[†] | 22 797 | | 24 282 | | 956 | | 1 239 | |
| 0 | | 9 784 (42.9) | | 9 889 (40.7) | | 304 (31.8) | | 370 (30.0) |
| 1 | | 8 537 (37.4) | | 9 051 (37.3) | | 369 (38.6) | | 463 (37.4) |
| 2 | | 3 157 (13.8) | | 3 590 (14.8) | | 180 (18.8) | | 235 (19.0) |
| 3 | | 859 (3.8) | | 1 114 (4.6) | | 54 (5.6) | | 112 (9.0) |
| ≥4 | | 463 (2.0) | | 638 (2.6) | | 49 (5.1) | | 59 (4.8) |
| Chronic hypertension[‡] | 22 797 | 182 (0.7) | 24 283 | 202 (0.7) | 966 | 23 (2.4) | 1 239 | 17 (1.4) |
| Smoking at first visit | 21 810 | | 23 139 | | 937 | | 1 206 | |
| No | | 21 094 (96.7) | | 22 309 (96.1) | | 880 (94.9) | | 1 144 (94.9) |
| 1–9 cig/day | | 597 (2.7) | | 668 (2.9) | | 37 (4.0) | | 51 (4.2) |
| ≥10 cig/day | | 119 (0.6) | | 162 (0.7) | | 10 (1.1) | | 11 (0.9) |
| Swedish snuff at first visit | 22 741 | 191 (0.8) | 24 085 | 236 (1.0) | 955 | 8 (0.8) | 1 232 | 18 (1.5) |
| Country of birth[§] | 22 790 | | 24 281 | | 956 | | 1 239 | |
| Sweden | | 15 592 (68.4) | | 16 444 (67.7) | | 588 (61.5) | | 670 (54.1) |
| Europe except Sweden | | 2 380 (10.4) | | 2 459 (10.1) | | 98 (10.2) | | 129 (10.4) |
| Middle East and North Africa | | 2 261 (9.9) | | 2 494 (10.3) | | 139 (14.5) | | 187 (15.1) |

(*Continued*)

**Table 1.** (Continued)

| | mITT population | | | | Subgroup discordant for definition of GDM* | | | |
| --- | --- | --- | --- | --- | --- | --- | --- | --- |
| | SWE-GDM criteria (*n* = 22 797) | | WHO-2013 criteria (*n* = 24 283) | | SWE-GDM criteria (*n* = 956) | | WHO-2013 criteria (*n* = 1 239) | |
| *North America and Caribbean* | | 101 (0.4) | | 114 (0.5) | | 3 (0.3) | | 4 (0.3) |
| *South and Central America* | | 303 (1.3) | | 288 (1.2) | | 14 (1.5) | | 16 (1.3) |
| *Africa* | | 1 256 (5.5) | | 1 562 (6.4) | | 77 (8.0) | | 135 (10.9) |
| *South East Asia* | | 290 (1.3) | | 318 (1.3) | | 18 (1.9) | | 54 (4.4) |
| *Western Pacific* | | 607 (2.7) | | 602 (2.5) | | 19 (2.0) | | 44 (3.5) |
| **Highest education, years** | 21 982 | | 23 512 | | 921 | | 1 190 | |
| *<9 (school education)* | | 678 (3.1) | | 858 (3.6) | | 49 (5.3) | | 85 (7.1) |
| *9 (school education)* | | 1 231 (5.6) | | 1 451 (6.2) | | 65 (7.1) | | 106 (8.9) |
| *10–11 (school education)* | | 1 530 (7.0) | | 1 688 (7.2) | | 79 (8.6) | | 139 (11.7) |
| *12 (school education)* | | 5 171 (23.5) | | 5 980 (25.4) | | 248 (26.9) | | 308 (25.9) |
| *<3 (college/university)* | | 3 455 (15.7) | | 3 498 (14.9) | | 145 (15.7) | | 164 (13.8) |
| *≥3 (college/university)* | | 9 678 (44.0) | | 9 746 (41.4) | | 323 (35.1) | | 378 (31.8) |
| *Doctor/licentiate degree* | | 239 (1.1) | | 291 (1.2) | | 12 (1.3) | | 10 (0.8) |
| **Plasma glucose in OGTT group, mmol/L, mean (SD)** | | | | | | | | |
| *Fasting* | 3 085 | 5.0 (0.8) | 3 733 | 5.0 (0.7) | 956 | 5.4 (0.4) | 1 239 | 5.5 (0.5) |
| *1-h* | 80 | 8.0 (2.1) | 3 042 | 8.2 (2.0) | 21 | 7.8 (1.4) | 1 085 | 9.3 (1.8) |
| *2-h* | 3 040 | 7.1 (2.0) | 3 659 | 7.0 (1.8) | 945 | 7.1 (1.3) | 1 210 | 8.0 (1.8) |
| **HbA1c in GDM group, mean (SD)** | 288 | 37.6 (6.6) | 746 | 34.9 (5.1) | | NA | 637 | 34.5 (4.2) |
| **Neonatal characteristics** | 22 738 | | 24 227 | | 951 | | 1 238 | |
| *Boy* | | 11 665 (51.3) | | 12 328 (51.0) | | 493 (51.8) | | 632 (51.0) |
| *Girl* | | 11 073 (48.7) | | 11 899 (49.1) | | 458 (48.2) | | 606 (49.0) |

Data are *n* (%) or median (IQR) unless stated otherwise.

*The cohort of women with fasting and 2-h plasma glucose cut off between the WHO-2013 and SWE-GDM criteria (fasting plasma glucose 5.1–6.9 and/or 2-h plasma glucose 8.5–8.8/8.9/9.9 mmol/L), untreated before and treated after the switch.

[†]Number of previous deliveries; stillbirths or live births.

[‡]Hypertension diagnosed before pregnancy or new onset hypertension with blood pressure ≥140/90 mmHg before gestational week 20.

[§]Grouped according to International Diabetes Federation Atlas except for having an extra category for Sweden.

BMI, body mass index; GDM, gestational diabetes mellitus; ITT, intention to treat; mITT, modified intention to treat; NA, not applicable; OGTT, oral glucose tolerance test; SD, standard deviation; WHO, World Health Organisation.

period (Table 1). In the SWE-GDM group, there were 994 (4.4%) pregnancies with missing data for potential confounders and in the WHO-2013 group 1 146 (4.7%), as reported in S8 Table.

There were no serious adverse maternal events reported during the CDC4G trial period. Adherence to the trial protocol is detailed in S2 Fig.

The prevalence of GDM increased from 2.6% (595 of 22 797 women) to 6.6% (1 591 of 24 283 women), producing an adjusted risk ratio (aRR) of 2.8 (95% CI 1.9 to 4.1, $p < 0.001$) following WHO-2013 criteria implementation in the mITT population. The prevalence of LGA in the mITT population remained unchanged after the switch, 11.4% before and 11.5% after, with aRR 0.97 (95% CI 0.91 to 1.02, $p = 0.26$), with no major heterogeneity between clusters and calendar time periods (S9 Table). LGA could not be classified for 164 of 47 080 (0.35%) neonates in the mITT population. Of the secondary neonatal outcomes in the mITT population, there was an association with increased cranial hemorrhage, respiratory distress, mechanical ventilation, and 5-min Apgar score <4 after the switch but a decreased risk in exploratory outcomes including mean birth weight, mean birth length, mean gestational age, macrosomia (≥4 500 g), and LGA >2 SD [16] (Tables 2 and S10).

In the mITT, there were missing values for 168 neonates in the SWE-GDM group and for 152 in the WHO-2013 group. In the subgroup, there were missing values for 12 neonates in the SWE-GDM group and for 4 in the WHO-2013 group.

For the other secondary maternal outcomes, there was a statistically significant reduced risk for gestational weight gain, shoulder dystocia, and perineal trauma (grades III and IV) after the switch in the mITT population (Table 3). The increased prevalence of GDM was associated with significantly increased numbers receiving GDM treatment, but no change in induction of labour. An inverse association with breastfeeding of infants at discharge was seen after the switch (S11 Table).

In the subgroup discordant for definition of GDM, the LGA prevalence was 273/956 (28.8%) before and 278/1 239 (22.5%) after aRR 0.87 (95% CI 0.75 to 1.01, $p = 0.076$). LGA data were missing for 11 of 2 195 (0.050%) pregnancies (Table 2). There was a statistically significant decrease in the association with the neonatal composite outcome (15/956 (1.6%) compared with 13/1 239 (1.0%); aRR 0.19 (95% CI 0.09 to 0.40, $p < 0.001$)) and an increase in neonatal hypoglycemia (glucose <2.6 mmol/L), but with no increase in need for intravenous therapy in neonates. There was a statistically significant decrease in the composite maternal outcome in the subgroup (144/956 (15.1%) compared to 141/1 239 (11.4%); with aRR 0.78 (95% CI 0.64 to 0.94, $p = 0.009$)).

There were statistically significant decreases in the aRR for the exploratory outcomes: macrosomia (≥4 500 g), LGA >2 SD), and secondary maternal outcomes: maternal composite postpartum hemorrhage, and in gestational weight gain (Tables 2 and 3). An increased risk for preeclampsia and gestational hypertension risk was seen after implementing the WHO-2013 criteria (Table 3). The switch was associated with reduced breastfeeding at discharge in the subgroup discordant for definition of GDM, which was also seen in the sensitivity analysis after multiple imputation of missing outcome data. The statistically significant inverse association with poorer self-reported health during pregnancy disappeared after imputation (S11 Table).

In the subgroup discordant for definition of GDM, the number needed to treat to avoid one composite neonatal outcome was 79 (95% CI 70 to 106), to avoid one composite maternal outcome 30 (95% CI 18 to 111), and to avoid one neonate born ≥4.5 kg 16 (95% CI 14 to 24) or severe LGA 26 (95% CI 16 to 151).

In the mPP population, the neonatal composite outcome could not be calculated due to few instances in the subgroup discordant for definition of GDM; otherwise, the results did not

**Table 2. Primary outcome and prespecified neonatal outcomes in the modified intention to treat population and subgroup discordant for definition of GDM*.**

| | mITT population | | | | Subgroup discordant for definition of GDM* | | | |
| --- | --- | --- | --- | --- | --- | --- | --- | --- |
| | SWE-GDM criteria (*n* = 22 797) | WHO-2013 criteria (*n* = 24 283) | WHO-2013 vs. SWE-GDM | | SWE-GDM criteria (*n* = 956) | WHO-2013 criteria (*n* = 1 239) | WHO-2013 vs. WE-GDM | |
| | | | Adjusted 1[†] RR (95% CI) | Adjusted 2[‡] RR (95% CI) | | | Adjusted 1[†] RR (95% CI) | Adjusted 2[‡] RR (95% CI) |
| **Primary outcome** | | | | | | | | |
| **Large for gestational age (>90th percentile)**[§][16] | 2 584 (11.4) | 2 790 (11.5) | 0.96 (0.92–1.01) $P^†$ = 0.11 | 0.97 (0.91–1.02) $P^‡$ = 0.26 | 273 (28.8) | 278 (22.5) | 0.83 (0.71–0.97) $P^†$ = 0.018 | 0.87 (0.75–1.01) $P^‡$ = 0.076 |
| **Secondary neonatal outcome** | | | | | | | | |
| **Composite neonatal outcome** | 288 (1.3) | 365 (1.5) | 1.14 (0.99–1.31) $P^†$ = 0.062 | 1.18 (0.99–1.39) $P^‡$ = 0.050 | 15 (1.6) | 13 (1.0) | 0.18 (0.08–0.38) $P^†$ < 0.001 | 0.19 (0.09–0.40) $P^‡$ < 0.001 |
| *Respiratory distress* | 148 (0.65) | 202 (0.83) | 1.43 (1.11–1.84) $P^†$ = 0.006 | 1.51 (1.12–2.02) $P^‡$ = 0.006 | 6 (0.63) | 6 (0.48) | NA | NA |
| *Spinal cord injury* | 0 (0.00) | 2 (0.01) | NA | NA | 0 (0.00) | 0 (0.00) | NA | NA |
| *Peripheral nerve/brachial plexus injury* | 16 (0.07) | 21 (0.09) | 1.24 (0.76–2.02) $P^†$ = 0.39 | NA | 1 (0.10) | 2 (0.16) | NA | NA |
| *Basal/depressed skull fracture* | 0 (0.0) | 0 (0.0) | NA | NA | 0 (0.0) | 0 (0.0) | NA | NA |
| *Clavicular fracture* | 31 (0.14) | 35 (0.14) | 0.76 (0.36–1.59) $P^†$ = 0.47 | 0.77 (0.36–1.66) $P^‡$ = 0.51 | 2 (0.21) | 1 (0.08) | NA | NA |
| *Long bone fracture* | 3 (0.01) | 0 (0.00) | NA | NA | 0 (0.0) | 0 (0.0) | NA | NA |
| *Cranial hemorrhage* | 37 (0.16) | 54 (0.22) | 1.65 (1.241–2.18) $P^†$ < 0.001 | 1.71 (1.26–2.33) $P^‡$ < 0.001 | 1 (0.10) | 0 (0.0) | NA | NA |
| *Stillbirth (≥22 gestational weeks) or neonatal death (day 0–28)* | 69 (0.30) | 83 (0.34) | 0.76 (0.54–1.08) $P^†$ = 0.13 | 0.76 (0.55–1.06) $P^‡$ = 0.11 | 5 (0.52) | 1 (0.08) | NA | NA |
| *Need of therapeutic cooling* | 20 (0.09) | 29 (0.12) | 1.44 (0.40–5.13) $P^†$ = 0.58 | NA | 0 (0.0) | 4 (0.32) | NA | NA |
| **Preterm birth (<37 weeks)** | 943 (4.1) | 1 144 (4.7) | 1.07 (0.96–1.19) $P^†$ = 0.23 | 1.08 (0.96–1.22) $P^‡$ = 0.21 | 39 (4.1) | 76 (6.1) | 1.18 (0.69–2.01) $P^†$ = 0.54 | 1.26 (0.72–2.22) $P^‡$ = 0.42 |
| **Small for gestational age (<10th percentile)**[§][16] | 2 525 (11.1) | 2 733 (11.3) | 0.98 (0.91–1.05) $P^†$ = 0.52 | 0.98 (0.90–1.06) $P^‡$ = 0.61 | 56 (5.9) | 104 (8.4) | 1.12 (0.88–1.44) $P^†$ = 0.35 | 1.13 (0.81–1.57) $P^‡$ = 0.46 |
| **Exploratory neonatal outcomes** | | | | | | | | |
| **Macrosomia (birth weight ≥4 500 g)**[§] | 679 (3.0) | 658 (2.7) | 0.78 (0.71–0.85) $P^†$ < 0.001 | 0.79 (0.72–0.87) $P^‡$ < 0.001 | 86 (9.1) | 48 (3.9) | 0.31 (0.19–0.53) $P^†$ < 0.001 | 0.32 (0.19–0.54) $P^‡$ < 0.001 |
| **Large for gestational age (>2 SD)**[§][16] | 1 019 (4.5) | 1 065 (4.4) | 0.30 (0.83–0.99) $P^†$ = 0.025 | 0.91 (0.84–0.99) $P^‡$ = 0.028 | 157 (16.6) | 141 (11.4) | 0.73 (0.57–0.93) $P^†$ = 0.009 | 0.77 (0.62–0.96) $P^‡$ = 0.020 |
| **Small for gestational age (<2 SD)**[§][16] | 659 (2.9) | 779 (3.2) | 1.04 (0.85–1.28) $P^†$ = 0.67 | 1.03 (0.82–1.29) $P^‡$ = 0.80 | 9 (0.95) | 36 (2.9) | 1.45 (0.75–2.81) $P^†$ = 0.27 | 1.52 (0.71–3.27) $P^‡$ = 0.28 |
| **Birth weight (g)**[§] | 3 528 (530) | 3 512 (543) | −22 (−35 to −10) $P^†$ < 0.001 | −21 (−37 to −5) $P^‡$ = 0.012 | 3 761 (552) | 3 572 (561) | −138 (−186 to 90) $P^†$ < 0.001 | −125 (−174 to −76) $P^‡$ < 0.001 |

(*Continued*)

**Table 2.** (Continued)

| | mITT population | | | | Subgroup discordant for definition of GDM* | | | |
| --- | --- | --- | --- | --- | --- | --- | --- | --- |
| | SWE-GDM criteria (*n* = 22 797) | WHO-2013 criteria (*n* = 24 283) | WHO-2013 vs. SWE-GDM | | SWE-GDM criteria (*n* = 956) | WHO-2013 criteria (*n* = 1 239) | WHO-2013 vs. WE-GDM | |
| | | | Adjusted 1[†] RR (95% CI) | Adjusted 2[‡] RR (95% CI) | | | Adjusted 1[†] RR (95% CI) | Adjusted 2[‡] RR (95% CI) |
| **Birth length (cm)[¶]** | 50.3 (2.4) | 50.2 (2.5) | −0.08 (−0.13 to −0.04) $P^†$ < 0.001 | −0.06 (−0.12 to −0.01) $P^‡$ = 0.019 | 50.9 (2.2) | 50.3 (2.4) | −0.30 (−0.48 to −0.12) $P^†$ = 0.001 | −0.31 (−0.42 to −0.21) $P^‡$ < 0.001 |
| **Gestational age (days)** | 278.1 (12.1) | 277.7 (12.7) | −0.47 (−0.62 to −0.33) $P^†$ < 0.001 | −0.46 (−0.63 to 0.28) $P^‡$ < 0.001 | 277.4 (10.7) | 274.8 (11.1) | −1.52 (−2.36 to −0.68) $P^†$ < 0.001 | −1.60 (−2.44 to −0.76) $P^‡$ < 0.001 |

Data are reported as *n* (%) or mean (SD).

*The cohort of women with fasting and 2-h plasma glucose cut off between the WHO-2013 criteria and SWE-GDM criteria (fasting plasma glucose 5.1–6.9 and/or 2-h plasma glucose 8.5–8.8/8.9/9.9 mmol/L), untreated before and treated after the switch.

[†]Analysed with multilevel mixed model adjusted for centre as random factor and period (January–March, April–June, July–September, and October–December) as fixed factor. Mixed Poisson model for binary outcomes (gives relative risk ratios as association measures), mixed multi-nominal for categorical outcomes (gives odds ratios as association measures), mixed linear model for continuous outcomes (gives mean differences as association measures), and mixed negative binomial model for count data (gives mean ratios as association measures).

[‡]Adjusted for mother's age modelled by a linear, squared, and cubic term, chronic hypertension, smoking, snuff, country of birth, and parity. Multiple imputation used for missing data on potential confounding variables.

[§]In the modified intention to treat population, there were missing values for 90 neonates in the SWE-GDM group and for 74 in the WHO-2013 group. In the subgroup, there were missing values for 9 neonates in the SWE-GDM group and for 2 in the WHO-2013 group.

CI, confidence interval; GDM, gestational diabetes mellitus; mITT, modified intention to treat; NA, not applicable; NICU, neonatal intensive care unit; RR, relative risk ratio; SD, standard deviation.

differ in significance between the mITT and mPP populations. Results for the mPP population are detailed in S12–S15 Tables.

In the analysis of the ITT population with pregnancies from centres that were excluded after randomisation due to protocol violation, number of pregnancies, and outcomes are reported in S16–S18 Tables.

## Discussion

In this SW-CRT of implementing the WHO-2013 diagnostic criteria in Sweden in a population screened using risk factors and repeated random plasma glucose measurements, there was a 2.5-fold increase in GDM diagnosis. The switch to the WHO-2013 criteria did not lead to a decrease in the primary outcome, LGA (>90th percentile) or composite neonatal or maternal outcomes in the mITT population. In the subgroup actually treated after the switch (based on fasting and 2-h values in the OGTT), no significant reduction in LGA (>90th percentile) was seen. However, there was a substantial reduction in the strength of the associations with both neonatal and maternal composite outcome; although more neonates were identified with hypoglycemia (glucose <2.6 mmol/L), without any associated increased need of intravenous glucose therapy. In the mITT population, there were adverse neonatal outcomes (respiratory distress, mechanical ventilation, cranial hemorrhage, and 5 minute Apgar score <4) that are unlikely to be a result of implementation of the WHO-2013 criteria since the sample size of the subgroup discordant for definition of GDM is small (only approximately 4% of the study population) and very few or no adverse outcomes were seen in the subgroup affected by the

**Table 3. Prespecified maternal outcomes in the modified intention to treat and in the subgroup discordant for definition of GDM.**

| | mITT population | | | | Subgroup discordant for definition of GDM* | | | |
|---|---|---|---|---|---|---|---|---|
| | SWE-GDM Criteria (*n* = 22 797) | WHO-2013 criteria (*n* = 24 283) | WHO-2013 vs. SWE-GDM | | SWE-GDM criteria (*n* = 956) | WHO-2013 criteria (*n* = 1 239) | WHO-2013 vs. SWE-GDM | |
| | | | Adjusted 1[†] RR (95% CI) | Adjusted 2[‡] RR (95% CI) | | | Adjusted 1[†] RR (95% CI) | Adjusted 2[‡] RR (95% CI) |
| **GDM prevalence** | 595 (2.6) | 1 591 (6.6) | 2.8 (1.9–4.2) $P^† < 0.001$ | 2.8 (1.9–4.1) $P^‡ < 0.001$ | 0 (0.0) | 1 239 (100) | NA | NA |
| **Composite maternal outcome** | 2 717 (11.9) | 2 727 (11.2) | 0.95 (0.89–1.02) $P^† = 0.15$ | 0.96 (0.91–1.02) $P^‡ = 0.17$ | 144 (15.1) | 141 (11.4) | 0.75 (0.59–0.95) $P^† = 0.018$ | 0.78 (0.64–0.94) $P^‡ = 0.009$ |
| *Shoulder dystocia* | 60 (0.26) | 42 (0.17) | 0.46 (0.26–0.81) $P^† = 0.007$ | 0.44 (0.27–0.70) $P^‡ < 0.001$ | 4 (0.42) | 3 (0.24) | NA | NA |
| *Perineal trauma (III and IV)* | 522 (2.3) | 502 (2.1) | 0.78 (0.68–0.88) $P^† < 0.001$ | 0.78 (0.68–0.89) $P^‡ < 0.001$ | 18 (1.9) | 23 (1.9) | 0.84 (0.51–1.39) $P^† = 0.51$ | NA |
| *Postpartum hemorrhage (≥1 000 ml)* | 2 269 (9.9) | 2 280 (9.4) | 0.99 (0.91–1.08) $P^† = 0.84$ | 1.00 (0.92–1.08) $P^‡ = 0.95$ | 128 (13.4) | 122 (9.8) | 0.74 (0.56–0.99) $P = ^†0.044$ | 0.77 (0.61–0.97) $P^‡ = 0.029$ |
| **Treatment during pregnancy** | | | | | | | | |
| *Diet only* | 269 (1.2) | 875 (3.6) | 3.82 (2.58–5.68) $P^† < 0.001$ | 3.80 (2.56–5.63) $P^‡ < 0.001$ | 10 (1.0) | 647 (52.2) | 53.4 (26.7–107) $P^† < 0.001$ | 54.2 (27.3–108) $P^‡ < 0.001$ |
| *Metformin only* | 204 (0.9) | 454 (1.9) | 2.06 (1.32–3.20) $P^† = 0.001$ | 2.01 (1.29–3.14) $P^‡ = 0.002$ | 0 (0.0) | 360 (29.1) | NA | NA |
| *Insulin only* | 37 (0.16) | 76 (0.31) | 1.48 (0.99–2.19) $P^† = 0.053$ | 1.46 (1.01–2.11) $P^‡ = 0.044$ | 0 (0.0) | 59 (4.8) | NA | NA |
| *Metformin and insulin* | 109 (0.48) | 200 (0.82) | 2.06 (1.16–3.66) $P^† = 0.013$ | 1.96 (1.10–3.51) $P^‡ = 0.022$ | 0 (0.0) | 166 (13.4) | NA | NA |
| **Gestational hypertension**[§] | 651 (2.9) | 798 (3.3) | 0.99 (0.70–1.40) $P^† = 0.95$ | 1.03 (0.72–1.47) $P^‡ = 0.86$ | 50 (5.4) | 81 (6.6) | 1.25 (0.95–1.64) $P^† = 0.11$ | 1.35 (1.01–1.80) $P^‡ = 0.045$ |
| **Preeclampsia**[¶] | 566 (2.5) | 676 (2.8) | 1.08 (0.87–1.34) $P^† = 0.50$ | 1.18 (0.92–1.49) $P^‡ = 0.19$ | 37 (3.9) | 66 (5.2) | 1.38 (1.02–1.85) $P^† = 0.033$ | 1.60 (1.14–2.24) $P^‡ = 0.007$ |
| **Gestational weight gain (kg)**[**] | 12.14 (5.6) | 12.14 (5.8) | −0.3 (−0.4 to −0.2) $P^† < 0.001$ | −0.3 (−0.4 to −0.2) $P^‡ < 0.001$ | 11.2 (6.9) | 9.4 (6.6) | −1.8 (−2.6 to −1.0) $P^† < 0.001$ | −1.6 (−2.4 to −0.9) $P^‡ < 0.001$ |
| **Cesarean section** | 4 027 (17.7) | 4 244 (17.5) | 1.02 (0.97–1.06) $P^† = 0.44$ | 1.02 (0.98–1.07) $P^‡ = 0.34$ | 255 (26.7) | 318 (25.7) | 0.95 (0.81–1.10) $P^† = 0.49$ | 0.92 (0.77–1.11) $P^‡ = 0.39$ |
| **Emergency cesarean section** | 2 980 (13.1) | 3 075 (12.7) | 1.02 (0.95–1.09) $P^† = 0.60$ | 1.01 (0.93–1.09) $P^‡ = 0.81$ | 187 (19.6) | 220 (17.8) | 0.91 (0.73–1.13) $P^† = 0.41$ | 0.90 (0.70–1.16) $P^‡ = 0.41$ |
| **Elective cesarean section** | 1 047 (4.6) | 1 169 (4.8) | 1.02 (0.91–1.14) $P^† = 0.73$ | 1.02 (0.92–1.14) $P^‡ = 0.67$ | 68 (7.1) | 98 (7.9) | 1.04 (0.70–1.55) $P^† = 0.85$ | 0.98 (0.62–1.53) $P^‡ = 0.92$ |

*(Continued)*

**Table 3.** (Continued)

| | mITT population | | | | Subgroup discordant for definition of GDM* | | | |
|---|---|---|---|---|---|---|---|---|
| | SWE-GDM Criteria ($n$ = 22 797) | WHO-2013 criteria ($n$ = 24 283) | WHO-2013 vs. SWE-GDM | | SWE-GDM criteria ($n$ = 956) | WHO-2013 criteria ($n$ = 1 239) | WHO-2013 vs. SWE-GDM | |
| | | | Adjusted 1[†] RR (95% CI) | Adjusted 2[‡] RR (95% CI) | | | Adjusted 1[†] RR (95% CI) | Adjusted 2[‡] RR (95% CI) |
| **Instrumental delivery** | 1 145 (5.0) | 1 225 (5.0) | 0.97 (0.88–1.06) $P^†$ = 0.50 | 0.97 (0.87–1.08) $P^‡$ = 0.58 | 31 (3.2) | 46 (3.7) | 1.06 (0.63–1.76) $P^†$ = 0.84 | 1.08 (0.65–1.80) $P^‡$ = 0.76 |

Data are $n$ (%) or mean (SD).

*The cohort of women with fasting and 2-h plasma glucose cut off between the WHO-2013 criteria and SWE-GDM criteria (fasting plasma glucose 5.1–6.9 and/or 2-h plasma glucose 8.5–8.8/8.9/9.9 mmol/L), untreated before and treated after the switch).

[†]Analysed with multilevel mixed model adjusted for centre as random factor and period (January–March, April–June, July–September, and October–December) as fixed factor. Mixed Poisson model for binary outcomes (gives relative risk ratios as association measures), mixed multi-nominal for categorical outcomes (gives odds ratios as association measures), mixed linear model for continuous outcomes (gives mean differences as association measures), and mixed negative binomial model for count data (gives mean ratios as association measures).

[‡]Adjusted for mother's age modelled by a linear, squared, and cubic term, chronic hypertension, smoking, snuff, country of birth, and parity. Multiple imputation used for missing data on potential confounding variables.

[§]Blood pressure ≥140/90 mmHg, measured 2 times with at least 4-h interval after gestational week 20.

Women with chronic hypertension diagnosis were excluded.

[¶]Blood pressure ≥140/90 mmHg and newly onset proteinuria ≥300 mg/24 h after gestational week 20.

**Adjusted for weight at first visit. In the mITT population, there were missing values for 1 715 women in the SWE-GDM group and for 2 203 in the WHO-2013 group. In the subgroup, there were missing values for 136 women in the SWE-GDM group and for 129 in the WHO-2013 group.

CI, confidence interval; GDM, gestational diabetes mellitus; mITT, modified intention to treat; NA, not applicable; RR, relative risk ratio; SD, standard deviation.

intervention. The reduced risk for the exploratory outcomes macrosomia (≥4.5 kg) and severe LGA in both the mITT and subgroup populations are clinically important outcomes relevant for decision-making. This reduction in birth weight is likely to be related to the benefits seen in maternal composite outcome (severe hemorrhage, perineal trauma, and shoulder dystocia). A decrease in breastfeeding at discharge was noted in both the mITT and the subgroup but may be non-generalisable due to the high rate of missing values for this self-reported outcome measure.

Two previous randomised controlled trials (RCTs) have studied the change from local guidelines to the WHO-2013 GDM criteria [7,23]. The most comparable RCT to our trial, the Gestational Diabetes Mellitus Trial of Diagnostic Detection Thresholds (GEMS), was conducted in New Zealand with a two-step OGTT screening [24] and reported no reduction in the primary outcome LGA in the total obstetric population but a reduction in their subgroup [7]. Differences in growth standards, population characteristics, screening methods, former diagnostic criteria, obstetric surveillance guidelines, and treatment targets make comparisons between the trials difficult and probably explain differences in measured outcomes. In the CDC4G trial, only women with risk factors for diabetes and high BMI were tested and treated, which is one major factor probably explaining some differences in outcomes. Furthermore, in the CDC4G trial, induction of labour was performed at 40+6 weeks' gestation at the latest for medically treated women and diet-treated women according to local guidelines up to 42+0 weeks'gestation, i.e., later than many other recommendations and guidelines [25].

Similar to the GEMS trial, we found an increased risk for neonatal hypoglycemia, likely due to surveillance bias from routine neonatal plasma glucose monitoring in neonates as previously shown [7]. However, identifying and treating more neonates with hypoglycemia might improve long-term neurocognitive outcomes [26].

The rate of preeclampsia differed between the trials within the subgroups discordant for definition of GDM, which was increased after the intervention in our trial but decreased in the GEMS trial. The increase in preeclampsia in our trial might be explained by surveillance bias but needs further evaluation. Furthermore, there were differences in breastfeeding rates at discharge between the studies: routines for supplementary feeding [27] might partly explain this difference. For example, in New Zealand, Dextrose gel was fully implemented during the period when the GEMS study was conducted [28]. In Sweden, Dextrose gel was recommended in the national guidelines for the first time in 2017 [29] and thus, not fully implemented during the CDC4G study. In addition, given the high proportion of missing values for self-reported breastfeeding in our study, these results should be treated with some caution.

Strengths include this being to the best of our knowledge, the first SW-CRT evaluating the WHO-2013 criteria enabling inclusion of approximately half of all deliveries in Sweden during 2018 in a real-world scenario with comprehensive data collection through national registries. The methodological complexities in the trial design included potential confounding with time and clusters, which were adjusted for in the analysis. The risk of selection bias is likely to be very low, as access to care is high (pregnancy care is free) and registers provided standardised medical information on all pregnancies, with coverage of >95% [15]: this makes the trial generalisable to a population screened by risk factors and also temporal trends in outcomes and/or possible residual confounding could be identified. The robustness of our data is further evident by the relatively unchanged risk after adjustments for various maternal characteristics, but residual confounding cannot be ruled out entirely. Also, the agreed treatment and surveillance guidelines that were implemented before starting, which is a major strength. We were able to implement the venous plasma sampling method across all the included clusters, and the Swedish national quality goals for glucose measurements were followed at all sites except one, including the use of fluoride citrate tubes for laboratory methods and quality assessment of patient near methods [30]. Even though 3 clusters were excluded from analysis, the study had adequate statistical power. To the best of our knowledge, this is the first major RCT of GDM criteria to use citrate to prevent ongoing glycolysis during the OGTT sampling making the glucose values more stable than using, e.g., fluoride alone [31].

Potential limitations of the trial are mainly due to the trial design. Although the planned sample size was exceeded, the power calculation was based on the assumption of the number of OGTTs generated by a universal one-step diagnostic approach [19].

We had to exclude 3 clusters (using one-step capillary OGTT screening) that were randomised, that were not able to change to venous OGTT as defined by the study protocol during the study period. This has however not introduced differential bias, since all pregnancies in all these clusters were excluded and no pregnancy from these centres was included in the mITT analysis [14]. Even though uniform treatment guidelines existed, it was impossible to control compliance with treatment and management strategies entirely. For the subgroup analysis, no comparison could be made based on the 1-h value, since the masking and introduction of a 1-h value in the OGTT was not possible to implement. As the duration of the study was only 1 year, we were unable to fully account for the seasonal variation which might include LGA and glucose values [32,33]. There was also a risk of chance positive findings due to multiple testing among the prespecified outcomes.

Concerns about implementing the WHO-2013 criteria have been raised previously [34], including increased resources and costs. The economic consequences have been analysed in conjunction with the CDC4G trial and will be reported separately. Whether the introduction and treatment of the WHO-2013 criteria result in long-term health benefits for

mother or child, needs to be evaluated in future follow-up studies in different populations [2–4,35].

Diagnosis of GDM increases the likelihood that women will attend postpartum follow-up programmes and may help to prevent future type 2 diabetes and cardiovascular disease.

The findings of the CDC4G trial must be placed within the wider discoveries in GDM research over the last 2 to 3 years. Like previous trials [7,36,37], the CDC4G trial found no benefit in reducing LGA defined as birth weight >90th percentile on a population level [16], but we could show a reduced risk for macrosomia ≥4.5 kg and LGA +2 SD [16] in the total pregnant population that most likely leads to the reduction in perineal trauma and shoulder dystocia. As with the GEMS study [7], a larger benefit occurred within the subgroup of women treated based on WHO-2013 criteria. This raises the question of why we treat GDM: for the benefit of the at-risk mother and baby or for the total obstetric population? Also, risk factor and random plasma glucose screening, with its lower sensitivity, denies many women GDM treatment [38,39] and the possibility to avoid adverse pregnancy outcomes. Beyond this, to the best of our knowledge, the CDC4G trial was the first to use citrate in a large trial for GDM, suggesting that treatment was actually commenced at a threshold below the HAPO cutoffs [31]. The implications of using more stable glucose sample handling, and other changes beyond thresholds, also need to be considered in any future changes in approach to diagnosing GDM. Finally, the recent findings of the TOBOGM study [40] suggest that new approaches to GDM screening and treatment need to include early detection and higher thresholds than the WHO-2013 criteria, since a raised risk for SGA was reported with treatment of lower levels of hyperglycemia (those treated between cutoffs for OR 1.75 and OR 2.0 from the HAPO study) [41] in early pregnancy in the TOBOGM study. With this new knowledge, it is obvious that the current screening and diagnostic approaches need to be reviewed: this represents an excellent opportunity for a coherent approach. We hope that this study, together with the other major RCTs and new scientific evidence, will contribute to the process that the Swedish National Board of Health and Welfare and other professional bodies in Sweden are working on to make changes in both screening, definition, and treatment of hyperglycemia during pregnancy. To date, to the best of our knowledge, no RCT has evaluated pregnancy outcomes after the implementation of the WHO-2013 criteria in a setting where universal one-step 75 g OGTT screening has been used. New technology and possible biological markers might be helpful in simplifying screening procedures and working towards a more individual approach in both identification and treatment of hyperglycemia during pregnancy for prevention of adverse outcomes in both the short and long term for mother and child.

Implementing the WHO-2013 diagnostic criteria for GDM in a risk factor-based screening setting did not reduce the risk of the primary outcome LGA (>90th percentile) in the total population or the subgroup affected by treatment. However, there was an associated reduction in adverse neonatal and maternal outcomes, with the largest effect in the subgroup of women whose OGTT results were discordant between the old and new criteria for the definition of GDM.

## Contributors

HB, SM, AM, and DS initially conceived and designed the study in collaboration with KB, HS, KH, ES, CU, UBW, VS, FA, AKW, ESL, SJ, and MP. AM independently performed the statistical analysis and SP undertook the data management. MdB wrote the first draft with input from HB. All co-authors contributed to trial implementation, discussion of the statistical analytical plan, interpretation of data, and critically revised and contributed to the final version of the manuscript. All authors had full access to all the data in the trial and had final responsibility for the decision to submit for publication. Local principal investigators are listed in S19 Table.

HB is the guarantor for the study and affirms that the manuscript is an honest, accurate, and transparent account of the study being reported; that no important aspects of the study have been omitted; and that any discrepancies from the study as originally planned (and, if relevant, registered) have been explained. The corresponding author attests that all listed authors meet authorship criteria and that no others meeting the criteria have been omitted.

## Patient and public involvement

There was no patient representative in the project group designing and performing the study. Client organisations will be involved in communicating the findings of the study to the general public.

## Supporting information

**S1 CONSORT Checklist. CONSORT checklist.**
(PDF)

**S1 Appendix. Research plan to ethics application.**
(PDF)

**S2 Appendix. Statistical analysis plan with corrections.**
(PDF)

**S1 Fig. Flow chart of the modified intention to treat population and subgroup discordant for definition of GDM in the CDC4G trial.**
(EPS)

**S2 Fig. Adherence to study protocol and checklists.**
(EPS)

**S1 Table. Intention to treat population in the CDC4G study according to cluster and number of births registered at delivery unit.**
(PDF)

**S2 Table. List of the CDC4G clusters, number of births, methods for screening, and diagnosing GDM.**
(PDF)

**S3 Table. Algorithm for starting (A), titration (B), maximum dose (C) for treatment (lifestyle advice, metformin, and insulin) in the CDC4G trial.**
(PDF)

**S4 Table. Target plasma glucose during pharmacological treatment.**
(PDF)

**S5 Table. The minimal requirements on obstetric surveillance during 2018 for participating centres.**
(PDF)

**S6 Table. Pregnancies included in the CDC4G trial based on gestational week, OGTT dates, and GDM status.**
(PDF)

**S7 Table. Coverage and certification level for data sources used in the CDC4G trial.**
(PDF)

**S8 Table. Comparing baseline characteristics in pregnancies with missing vs. complete potential confounding information by study groups in the modified intention to treat population.**
(PDF)

**S9 Table. Primary outcome in the modified intention to treat population and subgroup discordant for definition of GDM by period and cluster.**
(PDF)

**S10 Table. Prespecified secondary neonatal outcomes in the modified intention to treat population and subgroup discordant for definition of GDM.**
(PDF)

**S11 Table. Prespecified secondary maternal outcomes in the modified intention to treat population and subgroup discordant for definition GDM.**
(PDF)

**S13 Table. Primary and prespecified neonatal outcomes in the modified per protocol population and subgroup discordant for definition of GDM.**
(PDF)

**S16 Table. Characteristics of the intention to treat population.**
(PDF)

**S17 Table. Primary outcome and prespecified neonatal outcomes in the intention to treat population.**
(PDF)

**S18 Table. Prespecified maternal outcomes in the intention to treat.**
(PDF)

## Acknowledgments

We thank all personnel who contributed to collecting data and helped with the implementation and performing the study at all contributing centres. We want to specially thank U Hanson, A Ramnerö, I Nydahl, S Hogmark, A Esscher, L Rolfhamre, C Ragnarsson, B Pettersson, A-C Jonsson, A Storck, J Andersson, A Carlsson, A-L Fransson, A-M Wangel, K Kristensen, H Holmer, A Ahlsson, and S Brismar-Wendel. We are grateful for the support of the Swedish Network for clinical studies in obstetrics and gynecology, SNAKS (www.snaks.se).

## Author Contributions

**Conceptualization:** Scott Montgomery, David Simmons, Kerstin Berntorp, Stefan Jansson, Ulla-Britt Wennerholm, Anna-Karin Wikström, Helen Strevens, Fredrik Ahlsson, Verena Sengpiel, Erik Schwarcz, Elisabeth Storck-Lindholm, Carina Ursing, Karin Hildén, Helena Backman.

**Data curation:** Snehal Patil, Kerstin Petersson, Helena Backman.

**Formal analysis:** Anders Magnuson, Scott Montgomery, Helena Backman.

**Funding acquisition:** Maryam de Brun, Kerstin Berntorp, Verena Sengpiel, Elisabeth Storck-Lindholm, Helena Backman.

**Investigation:** Scott Montgomery, Snehal Patil, Ulla-Britt Wennerholm, Verena Sengpiel, Helena Backman.

**Methodology:** Anders Magnuson, Scott Montgomery, David Simmons, Kerstin Berntorp, Stefan Jansson, Ulla-Britt Wennerholm, Elisabeth Storck-Lindholm, Linda Ryen, Helena Backman.

**Project administration:** Maryam de Brun, Scott Montgomery, Anna-Karin Wikström, Martina Persson, Helena Backman.

**Resources:** Scott Montgomery, Kerstin Berntorp, Helen Strevens, Fredrik Ahlsson, Verena Sengpiel, Erik Schwarcz, Elisabeth Storck-Lindholm, Kerstin Petersson, Helena Backman.

**Software:** Anders Magnuson, Scott Montgomery, Snehal Patil.

**Supervision:** Scott Montgomery, David Simmons, Ulla-Britt Wennerholm, Erik Schwarcz, Elisabeth Storck-Lindholm, Martina Persson, Helena Backman.

**Validation:** Scott Montgomery, Kerstin Berntorp, Fredrik Ahlsson, Kerstin Petersson, Helena Backman.

**Visualization:** Maryam de Brun, Anders Magnuson, Ulla-Britt Wennerholm, Helena Backman.

**Writing – original draft:** Maryam de Brun, David Simmons, Helena Backman.

**Writing – review & editing:** Maryam de Brun, Anders Magnuson, Scott Montgomery, Snehal Patil, David Simmons, Kerstin Berntorp, Stefan Jansson, Ulla-Britt Wennerholm, Anna-Karin Wikström, Helen Strevens, Fredrik Ahlsson, Verena Sengpiel, Erik Schwarcz, Elisabeth Storck-Lindholm, Martina Persson, Kerstin Petersson, Linda Ryen, Carina Ursing, Karin Hildén, Helena Backman.

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
