## [Editor Report · Decision Letter 0]

21 Nov 2023

Dear Dr Backman, 

Thank you for submitting your manuscript entitled "Changing diagnostic criteria for gestational diabetes (CDC4G) in Sweden: A stepped wedge cluster randomised trial" for consideration by PLOS Medicine. Please accept our apologies for the delay in providing you with an editorial decision.

Your manuscript has now been evaluated by the PLOS Medicine editorial staff and I am writing to let you know that we would like to send your submission out for external peer review. 

During assessment of your manuscript, a number of differences were identified between your original, IRB-approved protocol and the results presented in your manuscript. We note that the inclusion criteria differ slightly, and that "women with pre-existing diabetes, previous gastric bypass surgery, or multifetal pregnancies were excluded" which was not specified in the original protocol. Please clarify this apparent difference in the main manuscript text. In addition, please ensure to report any outcomes that were not pre-specified as post-hoc, such as composite measure of respiratory distress, birth trauma, PPH, GDM treatment, and others. Please upload the revised manuscript and supplementary files as part of your resubmission. 

We would be grateful if you could re-submit your manuscript within two working days, i.e. by Nov 23 2023 11:59PM.

Feel free to email me at lgaynor@plos.org if you have any queries relating to your submission.

Kind regards,

Louise Gaynor-Brook, MBBS PhD

---

## [Decision Letter · Decision Letter 1]

18 Jan 2024

Dear Dr. Backman,

Thank you very much for submitting your manuscript "Changing diagnostic criteria for gestational diabetes (CDC4G) in Sweden using a stepped wedge cluster randomised trial" (PMEDICINE-D-23-03364R1) for consideration at PLOS Medicine. 

Your paper was discussed with an academic editor with relevant expertise and sent to independent reviewers, including a statistical reviewer. The reviews are appended at the bottom of this email and any accompanying reviewer attachments can be seen via the link below:

[LINK]

In light of these reviews, we will not be able to accept the manuscript for publication in the journal in its current form, but we would like to invite you to submit a revised version that addresses the reviewers' and editors' comments fully. You will appreciate that we cannot make a decision about publication until we have seen the revised manuscript and your response, and we expect to seek re-review by one or more of the reviewers. 

We hope to receive your revised manuscript by Feb 07 2024 11:59PM. Please email us (plosmedicine@plos.org) if you have any questions or concerns.

Please let me know if you have any comments, and we look forward to receiving your revised manuscript. 

Sincerely,

Richard Turner PhD, for Louise Gaynor-Brook, MBBS PhD

Consulting editor, PLOS Medicine

plosmedicine@plos.org

In order for the paper to be considered further at PLOS Medicine, it will need to comply with PLOS' data policy (https://journals.plos.org/plosmedicine/s/data-availability): i.e., anonymised study data should be made available in a publicly accessible repository, or a non-author contact provided for inquiries about access to the data. 

Our academic editor noted that a full ITT analysis should be presented alongside the mITT analysis. 

Where available, please quote p values alongside 95% CI. 

Noting comments from the referees, please provide the full trial protocol as an attachment, along with a statistical analysis plan. 

Comments from the reviewers:

*** Reviewer #1: 

The authors conducted a national stepped-wedge cluster RCT to determine whether the switch to the WHO-2023 diagnostic criteria in Sweden improves neonatal outcomes. The modified ITT analysis was conducted for eight clusters (14 delivery centres). The GDM prevalence increased after the switch to the WHO-2023 diagnostic criteria, though there was no change in the primary outcome of LGA. The subgroup analysis among women discordant between the two diagnostic criteria for GDM suggests the switch to the WHO-2023 criteria improved neonatal outcomes, including the statistically non-significant reduction of ~ 14% in the primary outcome of LGA (aRR: 0.86; 95% CI: 0.74 to 1.01).

The study was well-designed and executed; while the analysis was appropriately conducted. The manuscript is well written. Nevertheless, the authors might wish to address a couple of issues in the subsequent submission.

Major issues:

1. The Abstract should be revised as the Conclusions that implementing the WHO-2023 diagnostic criteria "decreased several adverse maternal and neonatal outcomes" were not supported by the findings. 

The analysis indicates the improved neonatal outcomes were found only in the subgroup of women discordant in the GDM diagnostic criteria, but not in the whole study population. Importantly, the switch to the WHO-2023 diagnostic criteria was associated with a significantly increased risk of several predefined neonatal outcomes (i.e., cranial haemorrhage, respiratory distress and 5-m Apgar score<4). The reduced risk of several exploratory neonatal outcomes (i.e., mean birthweight, birth length, mean gestational age, macrosomia and LGA� 2SD) associated with the intervention should be interpreted with caution given their exploratory nature.

2. Management of missing data should be described in detail.

- The authors might wish to add a sentence or two to describe whether a complete-case approach was conducted for missing data on the outcomes of interest. Given substantial magnitude of missing data on breastfeeding at discharge, an exploratory analysis using multiple imputation should be considered to its related discussion stronger. 

- It is reasonable to conduct multiple imputation for missing covariates. Please describe the magnitude and pattern of missing data on covariates. Importantly, given 10 imputed data sets generated for the missing potential confounding variables, the authors might wish to describe how the model parameters were calculated from these imputed data sets.

3. The unexpected increase in several predefined neonatal outcomes associated with the switch to the WHO-2023 diagnostic criteria is worth further discussion as the explanation "these findings were not seen in the subgroup" might be driven by small size of this subgroup (~ 4% of the study population in the primary analysis). 

Minor issues:

1. There were 11 clusters created from 17 delivery units, and a cluster randomisation was stratified by centre size. Details of delivery units, including their size by specified clusters should be described and presented, probably as a supplementary table.

2. Please complete this sentence "The switch was associated with p self reported health during pregnancy (good/very good), breastfeeding at discharge in the subgroup however 17.7-36.7% of the pregnancies were missing respectively (table S10)."

*** Reviewer #2: 

Statistical review

This paper reports a stepped wedge trial evaluating how the introduction of WHO-2013 criteria for diagnosing gestational diabetes affected maternal and neonatal outcomes.

The statistical methods used are appropriate and results are generally reported clearly. I had some comments on the reporting of the study:

1. Abstract: "After implementation of uniform clinical and laboratory guidelines, clusters were randomised to intervention (switch to WHO-2013 GDM criteria) each month from February to November 2018." - I would change to 'a number of clusters were randomised' or something similar to ensure it's clear that switching of clusters was during different months.

2. Abstract: adding a brief description of mITT definition would be useful.

3. Abstract: for the primary outcome I feel it's appropriate to include a p-value as the trial was powered to test the associated hypothesis.

4. Abstract: could it be made clearer whether the subgroup was pre-specified?

5. Abstract - "There were risk reductions": I would add 'significant.'

6. Abstract - "Implementing the WHO-2013 criteria in Sweden did not reduce the main": I would add 'significantly'. In the conclusion it is not clear that the 'decreased serious adverse … outcomes' is in the subgroup. Coming back to this after reading the paper, it might be referring to components of the composite outcome, which might be worth clarifying in the results of the abstract.

7. Page 7 "Pre-defined exclusion criteria according to SAP" - I did not follow how these could be pre-defined in a document published after the trial completed recruitment. I think a protocol would be needed to be included.

8. Page 9: it wasn't too clear to me how the randomisation was stratified by centre size - is this Gothenburg and Stockholm being treated separately or something in addition to this?

9. When I was invited to review the paper I was advised by the editor that they had some concerns about changes between the protocol and what was actually done, especially the change in analysis population and the addition of outcomes. They suggested a section that specifically mentions these changes and adds rationale for each - I would agree that this is important to add so the reader can judge how this affects the results.

10. I would recommend that results for all 11 clusters are presented as supplementary, although agree that the mITT analysis is likely the more robust analysis.

11. Table 2 and 3: I would personally recommend adding p-values which I feel add additional information to the confidence intervals and the overall conclusion of whether results were statistically significant (which I assume refers to p<0.05).

James Wason

*** Reviewer #3: 

Review for Plos Medicine: Changing diagnostic criteria for gestational diabetes (CDC4G) in Sweden: A stepped wedge cluster randomised trial, de Brun M et al. 

Thank you very much for asking me to review this well written manuscript. I would like to start by applauding the authors and the whole trial team for undertaking this enormous task. In this large step wedge cluster randomised trial investigating the effect of implementation of WHO-2013 GDM criteria, the authors conclude that changing criteria did not impact the primary outcome (LGA) although did improve a number of secondary outcomes, particularly amongst a subgroup discordant for diagnostic criteria. 

Specific comments (mainly clarification):

P8: Second paragraph - 'The primary analysis' … through to 'due to protocol violation' - Can you please go through this paragraph to clarify the concepts - it is not clear to me as is written:

For example, the sentence commencing 'The intention to treat population….' can you please define briefly (perhaps in brackets) which eligibility criteria needed to be fulfilled before intervention to enable inclusion in the mITT as opposed to ITT? In addition, the following sentence commencing 'The same analyses were performed in a pre-specified subgroup of women', through to the end of the brackets, is unclear - please clarify. I think it would also help if the subgroup was 'labelled' here for the first time, and throughout the text for example as you have labelled in the table - 'Subgroup discordant for diagnosis of GDM'.

Can you also emphasise here the detail that the 1hr glucose was not included for this analysis (and why).

P9 in the 'Procedures' section. Could you please define specifically here what the screening method to decide who will undergo OGTT diagnostic testing is in Sweden ('The screening method remained unchanged').

P12 - Secondary maternal outcomes: Please add in a little detail about which tools were used to assess self-reported health during, and after pregnancy, and satisfaction with childbirth.

P14 - 3rd paragraph of Results: In the text it isn't clear who the comparator groups are - presume from table higher BMI, parity etc is WHO 2013 vs SWE-GDM but not specified. Equally, when discussing the subgroup women, presume from table this is WHO-2013 vs SWE-GDM, but please specify in the text. Please also define 'the subgroup' here (e.g., as suggested above) and at the start of the second paragraph P15.

P15 - final sentence is unclear - what is 'p self-reported health during pregnancy' - think this might be a typo? And does the 17.7% refer to the questionnaire, and the 36.7% to breastfeeding info? Please clarify.

P17 - Comparison with other studies: This paragraph could perhaps be made more granular, including a discussion of the similarities as well as defining the differences in outcomes.

P18 - Please clarify (correct?) the first sentence here - 'routines for supplementary feeding may partly explain this difference'.

End P18/beginning P19 - with the exclusions, was the trial still adequately powered?

P19/20 - Implications for practice and future research: I don't come away from reading this with a sense of what the authors think is the best approach going forward based on the results of this study and other evidence, perhaps being more of a list than a discussion? Might it be possible to rewrite a little? For example, I wonder whether you feel able yet to comment as to whether Sweden plans to continue with the newly formed process or revert or what next steps will be? Is there a thought process as to what is the best approach for women who fall in the subgroup should reversion be a choice? (you might want to refer to the ongoing Dutch study precisely examining this subgroup - Tango DM, NTR7473)? Will it be feasible to follow up offspring (and mothers) of this cohort using Swedish registries? Can you be more precise in the abstract and again in this section with regards to reviewing current processes? Am I correct in thinking that you are suggesting moving away from the recommendation of using 1 'global' approach - were you perhaps thinking of a precision approach? Thoughts on how/what? Are you sugg

---

## [Decision Letter · Decision Letter 2]

13 May 2024

Dear Dr. Backman,

Thank you very much for re-submitting your manuscript "Changing diagnostic criteria for gestational diabetes (CDC4G) in Sweden through a stepped wedge cluster randomised trial" (PMEDICINE-D-23-03364R2) for review by PLOS Medicine.

I have discussed the paper with my colleagues and the academic editor and it was also seen again by three reviewers. I am pleased to say that provided the remaining editorial and production issues are dealt with we are planning to accept the paper for publication in the journal.

[LINK]

If you have any questions in the meantime, please contact me on lgaynor@plos.org.  

We look forward to receiving the revised manuscript by May 20 2024 11:59PM.   

Sincerely,

Louise Gaynor-Brook, MBBS PhD

Senior Editor 

PLOS Medicine

plosmedicine.org

Requests from Editors:

Thank you for your patience with a longer assessment process than we anticipated, and apologies for the delay in providing you with an editorial decision. 

General comments:

Please ensure that the full trial protocol is provided as a supplementary file, and please refer to this document in the main manuscript.

Please define all abbreviations at first use.

In RCTs, there is usually a distinction in the language in terms of causal vs associational for primary and secondary trial outcomes. Please use associational language for secondary outcomes.

Throughout the paper, please adapt reference call-outs to the following style: "... every year [1,2]." (noting the absence of spaces within the square brackets).

To help us extend the reach of your research, please provide any Twitter handle(s) that would be appropriate to tag, including your own, your coauthors’, your institution, funder, or lab.

Data availability:

Please confirm that the email address provided (foudatauttag@regionorebrolan.se) is not associated with any of the study authors. 

Title: Please revise your title according to PLOS Medicine's style. Please place the study design in the subtitle (ie, after a colon). We suggest “Changing diagnostic criteria for gestational diabetes (CDC4G) in Sweden: A stepped wedge cluster randomised trial” or similar

Abstract:

Please report your abstract according to CONSORT for abstracts (https://www.equator-network.org/reporting-guidelines/consort-abstracts), following the PLOS Medicine abstract structure (Background, Methods and Findings, Conclusions).

Abstract Methods and Findings:

Please ensure that all numbers presented in the abstract are present and identical to numbers presented in the main manuscript text.

Selected secondary outcomes should not be reported in the Abstract. Since not all secondary/pre-specified outcomes from this RCT are reported in this manuscript, please remove results relating to selected secondary and other pre-specified outcomes from the Abstract. 

Please include the important dependent variables that are adjusted for in the analyses.

In the last sentence of the Abstract Methods and Findings section, please describe 2-3 of the main limitations of the study's methodology.

Please include the study protocol document and analysis plan, with any amendments, as Supporting Information to be published with the manuscript if accepted.

Abstract Conclusions:

Please begin your Abstract Conclusions with "In this study, we observed ..." or similar, to summarize the main findings from your study, without overstating your conclusions. Please emphasize what is new and address the implications of your study, being careful to avoid assertions of primacy. 

Author Summary:

The Author Summary should immediately follow the Abstract in your revised manuscript. Please see our author guidelines for more information: https://journals.plos.org/plosmedicine/s/revising-your-manuscript#loc-author-summary

Please revise “higher glycaemic to the lower glycaemic” for clarity, referring perhaps to ‘plasma glucose’ instead

Please define RCT at first use within the Author Summary.

Please revise “analysis was of both in a modified intention to treat (mITT) population (all pregnancies) as well as in a subgroup actually effected by the switch” for clarity - we suggest “analysis was conducted for all pregnancies, as well as in a subgroup actually affected by the switch”

Please define LGA at first use within the Author Summary.

Please remove all results relating to selected secondary outcomes; please only report results relating to the primary outcome (LGA). 

In the final bullet point of ‘What Do These Findings Mean?’, please describe the main limitations of the study in non-technical language.

Methods:

Please ensure that the full trial protocol is provided as a supplementary file, and refer to this early in the Methods section. Any changes in your analysis - including those made in response to peer review comments - should be identified as such in the Methods section of the paper, with rationale. 

Please include the completed CONSORT checklist as Supporting Information. Please add the following statement, or similar, to the Methods: "This study is reported as per the Consolidated Standards of Reporting Trials (CONSORT) extension for the stepped wedge cluster randomised trial (SW-CRT) (S1 Checklist)." The guideline can be found here: https://www.equator-network.org/reporting-guidelines/consort-cluster/ When completing the checklist, please use section and paragraph numbers, rather than page numbers which will likely no longer correspond to the appropriate sections after copy-editing.

Line 185 - please define SW-CRT at first use 

Line 212 - The SAP only refers to ITT; please define the difference between ITT and mITT in your analyses.

If results are presented for outcomes which were not prespecified in the SAP, please indicate that they were post hoc and explain why they were added. Post hoc comparisons should be presented as hypothesis generating rather than conclusive.

Results: 

Line 391 - please define mPP at first use. 

Line 429 - Please indicate which factors are adjusted for, as table S9 only provides RR values.

Discussion:

Please remove all subheadings within your Discussion e.g. Comparisons with other studies

Lines 528, 658 - please revise to reflect more associational language for secondary outcomes. 

Lines 596, 633, 647 - please temper assertions of primacy by adding ‘to the best of our knowledge’ or similar.

Please remove the information on competing interests, funding and data sharing from the

end of the main text. In the event of publication, this information will appear in the article

metadata, via entries in the submission form.

Tables:

Please ensure to define all abbreviations used in each table in the respective table legend. 

Tables 2, 3, S10, S11 - please specify in the table legend which factors are adjusted for in ‘Adjusted 1’ results. When a p value is given, please specify the statistical test used to determine it in the table legend.

References:

Please ensure that journal name abbreviations match those found in the National Center for Biotechnology Information (NCBI) databases (http://www.ncbi.nlm.nih.gov/nlmcatalog/journals), and are appropriately formatted and capitalised.

Please also see https://journals.plos.org/plosmedicine/s/submission-guidelines#loc-references for further details on reference formatting. 

Where website addresses are cited, please specify the date of access. 

Comments from Reviewers:

Reviewer #1: I would like to thank the authors for their efforts to address my questions. I have no additional comments to make.

Reviewer #2: Thank you to the authors for addressing my previous comments well. The only minor issue I had was on my initial review's point 1. Personally I would recommend adding 'a number of' before 'clusters' but this can be improved at the post-acceptance proofing stage. I had no other issues to raise.

Reviewer #3: Thank you for responding to each of our comments with such care. I have no further comments to add.

[LINK]

---

## [Editor Report · Decision Letter 3]

29 May 2024

Dear Dr Backman, 

On behalf of my colleagues and the Academic Editor, Prof. Jenny Myers, I am pleased to inform you that we have agreed to publish your manuscript "Changing diagnostic criteria for gestational diabetes (CDC4G) in Sweden: a stepped wedge cluster randomised trial" (PMEDICINE-D-23-03364R3) in PLOS Medicine.

Before your manuscript can be formally accepted you will need to complete some final editorial and formatting changes, which you will receive in a follow up email. Please be aware that it may take several days for you to receive this email; during this time no action is required by you. Once you have received these formatting requests, please note that your manuscript will not be scheduled for publication until you have made the required changes.

PRESS

Sincerely, 

Louise Gaynor-Brook, MBBS PhD 

Senior Editor 

PLOS Medicine

lgaynor@plos.org